# Self-Inflicted Burns: A Comparative Study in a Spanish Sample

**DOI:** 10.3390/ebj6010008

**Published:** 2025-02-17

**Authors:** Sara Guila Fidel-Kinori, Vicente García-Sánchez, Maria Sonsoles Cepeda-Diez, Carmina Castellano-Tejedor, Josep Antoni Ramos-Quiroga, Joan Pere Barret-Nerín

**Affiliations:** 1Campus Vall d’Hebron, University Hospital Vall d’Hebron, 08035 Barcelona, Spain; vicente.garcia@vallhebron.cat (V.G.-S.); mariasonsoles.cepeda@vallhebron.cat (M.S.C.-D.); ninacastej@yahoo.es (C.C.-T.); antoni.ramos@vallhebron.cat (J.A.R.-Q.); juanpedro.barret@uab.cat (J.P.B.-N.); 2Vall d’Hebron Research Institute, 08035 Barcelona, Spain; 3Department of Psychology and Education Sciences, Universitat Oberta de Catalunya (UOC), 08018 Barcelona, Spain; 4Department of Psychiatry, Autonomous University of Barcelona, 08192 Bellaterra, Spain; 5Hospital Sant Joan de Déu de Barcelona, 08950 Barcelona, Spain

**Keywords:** self-inflicted burns, suicide, comparative study

## Abstract

Background: In 1994, the first Spanish study on patients with self-inflicted burns (SIB) was published, showing a prototypical profile of a patient with SIB: adult male, unmarried and, in 75% of the cases, with a psychiatric background. In addition, SIB accounted for 1.98% of the total admissions in a Burns Unit between 1983 and 1991, a lower percentage than other European studies. The present study aims to replicate this work, updating this profile and comparing it with the current profile. Methods: We compared the clinical and socio-demographic characteristics of 67 patients admitted during 1983–1991 (Study I) with those of 36 patients admitted during 2010–2015 (Study II). Results: It was observed that the percentage of patients with SIB admitted to the Burns Unit was lower in Study II than in Study I (1.45% vs. 1.98%). Significant age differences were identified (*t*_(101)_ = −2.074, *p* = 0.041, 95% CI [−11.739, −0.261]). Similarly, there were statistically significant differences in several clinical characteristics, such as psychiatric history (X^2^ = 11.591, *p* = 0.001), the occurrence of previous autolytic attempts (X^2^ = 7.714, *p* = 0.007), the place where the incident occurred (X^2^ = 11.647, *p* = 0.020), the etiology of the burn (X^2^ = 13.142, *p* = 0.004), and triggers (X^2^ = 6.420, *p* = 0.036). Conclusions: Several differences have arisen between the two studies, mainly related to the specific characteristics of SIB (e.g., etiology, triggering cause, and place of the incident), possibly attributable to the social changes that have occurred in the last 20 years. These results will add to our knowledge and will stress various precipitating factors that may lead to SIB, with the final goal of designing preventive strategies.

## 1. Introduction

According to the World Health Organization (WHO), suicide is the leading cause of violent death worldwide, and for every completed suicide, there are several autolytic attempts, whose values differ according to the country [1].

Within the population attended to in Specialized Burns Units (SBUs), there is a specific group of patients admitted due to self-harm behaviors produced by fire and/or by means of explosive mechanisms [2,3]. According to the different studies, the most frequently used mechanism in SIB is flame [4,5,6,7,8] and, because of this act, the burned body surface can vary between 16.5% and 40.4% [3,4,9,10,11,12]. Self-inflicted burns (SIB) are defined as injuries caused by deliberate self-harm involving thermal agents, resulting in hospital admission. This definition follows the criteria used in Study I by García-Sanchez et al. [6], ensuring consistency with the original research parameters. It is important to clarify that large parts of the studies reviewed use the concept of attempted suicide by fire, equivalent to SIB. This work uses the term SIB due to its broader and more inclusive nature.

Despite the severity of these behaviors, SIB are relatively scarce and vary widely according to the countries or regions. In this sense, people who commit SIB as an autolytic attempt may present a huge variety of motives and reasons, in many cases linked to diverse socio-cultural causes [13,14]. In the context of mental health and suicidology, “autolytic attempt” refers to a self-directed act with intent to cause one’s death. It encompasses acts with varying levels of lethality and salvageability, reflecting the individual’s underlying suicidal intent. Lalöe [3], in a review of the literature on SIB based on 55 studies from different geographical areas, identified three differential patterns or groups according to each region. In the so-called Western and Middle Eastern countries, the causes were mainly related to psychiatric disorders. A second group consisted of countries such as India, Sri Lanka, Papua New Guinea, and Zimbabwe, where the causes described were mainly related to personal factors (i.e., being exposed to or having suffered several stressful life events). Finally, the third group was represented by countries such as South Korea and again, certain regions of India. In these regions, SIB were often linked to political reasons. Focusing on European countries, it is observed that the prevalence of SIB is lower than in other areas of the world. For instance, Palmu et al. [11], in their study with Finnish population, documented a prevalence of 5.7%. Similarly, the prevalence of SIB in Greece was around 4% [4], between 1.16 and 3.5% in the United Kingdom [12,15], 4.2% in Australia [7], or 4% in Italy [5]. Despite the variation according to the country and the time of study, in Catalonia, a relatively low prevalence of 1.98% stands out, as published by García-Sánchez et al. [6].

According to publications on European population, the socio-demographic characteristics of this population committing SIB show a higher prevalence of males, with an average age of 30–40 years [10,11,12], in line with the findings of García-Sánchez et al. with the Catalan population [6]. However, in other countries, different results have been found in terms of gender, with an unclear pattern or with significant differences in terms of the prevalence of males and females. This is the case of the studies carried out in Finland [11], Bulgaria [14], and Italy [5], or even in studies carried out in Greece, where a majority of women committing SIB has been observed [4]. However, all the reviewed studies agree in identifying a more prevalent profile at risk of committing SIB, comprising a majority of single or divorced persons [4] without paid employment at the time of the study [6,11]. In addition, several studies underline that this at-risk population is often characterized by a poor social support network [15].

As indicated above, another relevant aspect identified in these patients admitted to Burns Units due to SIB is the existence of a psychiatric history, comprising more than 80% of the cases, according to some studies [11,12]. Despite the heterogeneity, the most frequently identified diagnosis is psychotic disorder [8,11,14,16], followed by substance abuse [17], and/or affective disorders [11,18,19]. It has also been observed that many of these patients had committed previous autolytic attempts [11,12,17], not necessarily with fire. In this sense, Titscher et al. [20] reviewed admissions due to SIB in the Burns Unit of Vienna from 1994 to 2005. They analyzed the clinical and socio-demographic variables and presented a classification according to the triggering factor of the self-injurious behavior. The classifications were as follows: (1) typical, when the patient presents a previous psychiatric history with other autolytic or self-harm attempts; (2) delirious, when patients perform the act under the influence of a psychotic disorder or the effects of substance abuse; and (3) reactive, when wounds are a way to avoid negative emotions produced by intensely stressful events. Other authors following this same line of research have used a binary classification with these patients: (1) those who perform these acts under the idea of self-harm and (2) those who make a clear attempt at autolysis [14].

Regarding the evolution of these patients, the mortality rate is diverse, and the data are not yet conclusive. Some authors highlight that the mortality risk is the same as in patients with self-inflicted injuries due to other causes [8,21,22,23,24].

Considering that this sample of patients is scarce, as described above, and results describing their specific characteristics are still far from conclusive, this work aims to compare data from a study covering the period 1983–1991 [6], which we call Study I, with the current data in the same reference center in a sample of patients admitted for SIB during the period 2010–2015, which we call Study II.

This former study was published in 1994, and was the first study on patients admitted for SIB in a Burns Unit in Barcelona. The results showed that patients committing SIB accounted for 1.98% of the total admissions in the Unit, and comprised predominantly unmarried males with a previous psychiatric history. The most relevant fact was that the prevalence of this population was much lower than that of other similar European studies.

We hypothesized that changes occurring during the last twenty years in Catalonia—a significant increase in immigration (currently, >14% non-native population, compared to <1% in previous decades), the economic crisis (since 2008, severe consequences for living conditions), and the technological changes, especially in information and communication technologies (ICTs)—may produce significant differences in the profile of patients admitted to the Burns Unit after an SIB. If this is the case, the results will allow us to continue to advance toward a correct characterization of these patients, providing information for a better clinical approach and identifying their specific needs for medical and psychosocial care.

## 2. Materials and Methods

**Design:** Like García et al.’s [6] research, this is a cross-sectional retrospective naturalistic study.

**Setting:** This is a comparative study of two samples from the same center but differentiated in two periods. The first study (Study I) comprises the period from 1983 to 1991, and the second study (Study II) comprises the period from 2010 to 2015. The Burns Unit in Barcelona is currently receiving patients from an area of influence of more than 8,500,000, including the local population, the Balearic Islands, and the Andorra Princedom. It is the second largest Burns Unit in Spain, located in a tertiary general reference hospital in Barcelona. This Unit has a multidisciplinary health team and performs highly specialized burn treatments.

**Patients:** The total number of inpatients in the Burns Unit during the years of Study I was 3.371 (N = 67; 1.98% of the total admitted patients with SIB during this period). Concerning Study II, carried out between 2010 and 2015, the total number of patients admitted to the Burns Unit was 2467, and the final sample of patients with SIB consisted of 36 (1.45% of total admitted patients meeting the inclusion and exclusion criteria). The total number of patients from both studies was 103 (78 males and 25 females).

**Inclusion and exclusion criteria:** Patients included in the study met the following criteria: (1) males and females, (2) age ≥ 18 years old, and (3) hospitalized because of an autolytic attempt with fire or SIB.

**Materials/Variables:** The following socio-demographics, clinical, and medical–surgical variables were collected:Socio-demographics: gender; age; marital status.Clinical variables: place of the incident (home, psychiatric hospital, jail, public area, others); method used to cause the burn (flame, gas, electrocution, others); psychiatric history (yes/no); previous suicide attempts (yes/no); trigger of SIB (psychopathology, psychosocial).Medical–surgical evolution: length of hospital stay (in days), mortality.

**Procedure:** Data from consecutive patients admitted to the specialized Burns Unit who met all the admission criteria due to SIB were recorded consecutively. As sources, we consulted the Emergency department’s reports and the interviews with the patient and/or their direct relatives, retrieving additional information not included in medical records.

**Ethical issues:** This research complies with internal ethical requirements and the Helsinki Declaration of 1975 and its subsequent amendments. As no direct intervention with patients was performed, no additional ethical standards were necessary. The anonymity and confidentiality of the data were guaranteed to avoid participants’ identification.

**Statistical analyses:** Descriptive analyses were carried out and the mean frequency of the variables included in the first study published in 1993 was determined. When comparing the samples’ demographic, clinical, and burn injury characteristics, Chi-square tests, Fisher’s exact test, two-sample *t*-tests, and Wilcoxon rank sum tests were used to determine the statistical significance of group differences (categorical and continuous variables). The significance level was set at *p* < 0.05, and confidence intervals were indicated. Data were analyzed with the SPSS 20 program.

This comparative and exploratory study was the first step in a large and integral study accepted by the Ethical Committee of the Vall d’Hebron University Hospital PR(AG)628/2020.

## 3. Results

Descriptive results from the two samples (Study I and Study II) are summarized in Table 1.

Significant age differences between studies were observed (t_(101)_ = −2.074, *p* = 0.041, 95% CI [−11.739, −0.261]), place of the incident (X^2^ = 11.647, *p* = 0.020), method to produce the burn (X^2^ = 13.142, *p* = 0.004), psychiatric history (X^2^ = 11.591, *p* = 0.001), previous suicide attempts (X^2^ = 7.714, *p* = 0.007), and precipitants (X^2^ = 6.420, *p* = 0.036).

Thus, in Study I, the mean age was significantly lower than in Study II (38 vs. 44), and the home was the most frequently chosen place to commit the SIB, with lower rates in other spaces (67% vs. 53%). In this sense, an increase in public areas was observed (33%) in Study II. Similarly, despite flame being the most common method used in both studies, Study II revealed a clearer preference for this method (94%), compared to 66% for flame and 28% for gas in Study I. Moreover, psychiatric history and previous suicide attempts were significantly higher in Study II (92% vs. 75% and 47% vs. 21%, respectively). Finally, the etiology or main trigger for committing SIB was psychopathology in Study I (73%), whereas in Study II, psychopathology (53%) and psychosocial issues (44%) were balanced.

## 4. Discussion

The objective of this study was to compare the data of patients admitted for SIB in a specialized Burns Unit under the same admission criteria at two different times. That is, in Study I, between 1983 and 1991, and in Study II, between 2010 and 2015. Considering the results of Study I, we examined the changes in these patients’ profiles to determine their possible relationship with the relevant socio-cultural and economic changes that had occurred throughout the period of analysis.

Methodologically, a replication of the initial design was performed; therefore, in the analysis, only the variables considered in Study I were included, although, in Study II, other variables were also measured, which will serve as the basis for subsequent studies.

In Study II, people with SIB accounted for 1.45% of the admissions in the Burns Unit compared to 1.98% in Study I. Moreover, significant differences were identified in terms of age, psychiatric history, the occurrence of previous autolytic attempts, the place of the incident, the method to produce the burn, and the precipitating factors.

Concerning the admission rate for SIB in both studies, we observed that the percentages remain relatively low in absolute terms. This aligns with what has already been documented in other European studies, reporting a mean admission rate for SIB of around 4% [4,5,7,11,12]. The only exception is the study in the United Kingdom, which reported 2% [21], similar to the results obtained in this research. These results can be assessed from different perspectives. On the one hand, it must be considered that this is not an epidemiological study. Thus, it focuses on the clinical data collected at the moment of admission at the Burns Unit. This means that some SIB could have been overlooked, and other admission causes may have been noted. Moreover, the study was carried out with a clinical sample admitted to a specialized Burns Unit from a tertiary general hospital. Thus, populations not requiring admission to an emergency unit and attended to in other health units (e.g., outpatient consultation) were not included in this study. Patients deceased before referral to the corresponding health unit were also excluded.

Another aspect that should be considered is that, in 2016, the Aquas Agency published a study reporting rates of mortality due to suicide and suicide attempts in Catalonia, highlighting that the prevalence of such suicide behaviors was one of the lowest in the country, even lower than the rates reported by other European countries [25]. Therefore, it could be argued that the low rate of SIB is congruent with the low rate of suicides and suicide attempts in this territory. The effect of different cultural and environmental aspects may underlie this prevalence. Such differences specifically linked to SIB according to region have not yet been studied.

Regarding the information obtained in the comparison of socio-demographic data, especially in terms of gender, age, and marital status, in both periods, a greater proportion of males was observed, similar to other studies conducted in the Western population [2,3,17,20]. Despite this profile, other studies conducted in Finland [11], Bulgaria [14], and Italy [5] found no statistically significant gender differences when analyzing SIB. Likewise, in a study carried out with the Greek population [4], a greater proportion of women committing autolytic attempts with fire was found. Despite this predominance of males, no statistically significant gender differences were found in our study.

The mean age of the population committing SIB was also similar to that indicated in other studies in the European population [10,11,12]. However, there were statistically significant differences between the time periods of our two studies, revealing a significantly higher age in the population of Study II (38 vs. 44 years).

This work has also identified, as in other publications, a greater presence of single or divorced individuals [4,20]. We think this could be related to the person’s living conditions, suggesting a poor or insufficient social support network. Thus, most patients in Study II (single people) admitted to some psychosocial aspects (almost as important as psychopathology) as the precipitating motives for the attempt. The published literature indicates that these acts are habitually carried out by single people without a partner [26], those who are unemployed, and those with a low income [4,8,11,12].

In short, this profile coincides with people at risk of social exclusion. This coincides with other studies on suicide in the general population that emphasize that culture, job satisfaction, and economic and civil status are key factors related to the increase/decrease in the risk of committing suicide behaviors [3,20]. Thus, the profile of people who attempt suicide might differ depending on whether they are from a country with a high or low socio-economic level [13,24].

Regarding the variables that were not included in Study I but that might be related to the above-mentioned hypothesis stating that the economic and social changes in the last twenty years could explain the changes in the results found, we must consider the increase in the rate of non-native population admitted to the Burns Unit. Of the 36 patients, 25% of them were foreign non-native (*n* = 9). The term “foreign” refers to individuals who were not born in one specific country but have migrated there, often facing socio-economic challenges and limited support networks. In our study, the term “foreign” refers specifically to individuals born outside of Spain who immigrated to the country. This distinction is crucial, as research shows that immigrant populations often experience unique socio-economic and psychosocial stressors, including limited access to social support networks, economic difficulties, and challenges in cultural adaptation, all of which can influence health behaviors and self-harm risk. The findings of our study suggest that these factors may play a significant role in the etiology and characteristics of self-inflicted burns (SIB) among non-native individuals, as evidenced by their increased representation within our sample compared to historical data. Further research is needed to explore the specific mechanisms by which these social determinants impact SIB behavior and to design culturally sensitive interventions. Twenty years ago, the presence of an immigrant population was not relevant. The significant immigration flow since 2001 has introduced demographic changes in the population. According to the latest studies, in Catalonia, the non-native population is 14%. This clinical subgroup of the current sample is almost precisely double the overall ratio (https://www.idescat.cat/poblacioestrangera/?b=0&lang=es) (accessed on 12 July 2024).

Another noteworthy fact in this comparison is that, in Study II, there is an increase in the percentage of patients with a psychiatric history and with more previous suicide attempts compared to the data identified in the patients of Study I. There are several possible interpretations of this difference; among them, the fact that the current sample was collected by mental health specialists, who are more sensitive to the identification of this information, could have played a relevant role. Also, concerning the changes in data collection in healthcare centers, the computerization of medical records allows for more agile and clearer access to patients’ history in various care settings.

In Study I, most SIB occurred in private places like the home. In contrast, in Study II, although private places are still the preferred scenarios for committing such acts, public spaces have started to increase. Similarly, fewer incidents occurred in psychiatric admission centers (decreasing from 12% to 3%). These changes could have different causes. First, one might think that many of these SIB may be due to grievances (demand due to unfair dismissal, hopelessness due to socio-economic causes, etc.). These qualitative aspects collected during the interview with patients make us think that it is likely that public spaces are chosen as the most propitious scenarios to carry out these acts, which, to some extent, could be called “acts of protest.” On the other hand, psychiatric centers and prisons have modified their control and surveillance measures, which have improved the protection and security conditions offered to patients.

Regarding the method of committing self-harm, in both studies, flame was still the most frequent. This fact is consistent with the existing literature on the subject [3,4,11,12,14]. Thus, the use and nature of fire accelerants vary according to their availability in different countries. Kerosene and petrol are the most frequently used substances due their ease of access and cost.

Concerning the precipitating causes of the incident, the study by García et al. [6] found that psychopathology was a causal element, particularly a condition of “psychiatric destabilization.” In contrast, in Study II, an increase in psychosocial factors (work and economic problems, among others) was observed to be the main trigger for SIB. This is consistent with the causes associated with the increase or decrease in autolytic attempts, reported by several epidemiological studies. In this sense, the role of psychosocial variables as a risk for [27,28,29,30] or protective factor against [31] committing this type of autolytic attempt has often been highlighted. Likewise, this aligns with other studies stressing the importance of these factors in suicide by SIB in particular [2,4]. This may be relevant because of the possible effects produced by the changes over the past three decades, such as the economic crisis, immigration, technological changes, and political and employment instability [5,24,32].

Significant differences are also observed in the rate of the admitted patients’ psychiatric background and the place of the incident, distributed between home and public spaces, as well as the type of triggers of the act, which are changing from psychopathological to psychopathological and psychosocial aspects at present. In addition, regarding socio-economic health determinants, continued evidence from the scientific literature reveals an important increase in these external precipitators of SIB behavior [26,32].

Concerning the burnt body area, in Study I, one-third of the patients had burns on around 10–30% of their body or even <10%, whereas the remaining two-thirds were divided into burns of greater surface area. In contrast, in Study II, more than 60% of patients had burns on <30% of their body. These data also agree with similar studies [2,5,9,10,11,12]. However, the observed differences did not reach statistical significance in the present research.

In addition, despite not finding statistically significant differences, a shorter hospitalization time and fewer deaths were observed in Study II than in Study I. These are key factors indicating a good prognosis and satisfactory incident resolution. These data can also be interpreted according to the improvement in treatments and the general therapeutic approach to this type of patient over the years.

Finally, at the clinical level, as expected by the advances in healthcare, the time of hospitalization and mortality rates of these patients have been reduced. This raises questions about the survivors’ specific needs, the longitudinal evolution of their physical sequelae, and their psychopathological state and resolution after committing SIB.

Several limitations of this research should be taken into account when analyzing results. The data obtained are specific to patients admitted under the diagnostic criteria of SIB, making this research a quasi-epidemiological study for this geographic region. It is known that admitted patients to Burns Units are not the only ones committing SIB. Depending on the severity of the case, some patients may end up receiving ambulatory treatment. Consequently, it must be borne in mind that this is a descriptive study of the patients admitted to a Burns Unit, and not an epidemiological investigation. Additionally, there was a difference in the time span of the data collection. Study I collected data for nine years, while Study II only collected data for five years. Another limitation concerns the variables compared. In order to replicate Study I, only a few variables were analyzed in Study II. Due to the advances in this field, we need to extend these patients’ characterization with more relevant variables such as substance use or socio-economic level (some of these variables were collected in Study II but were not included in this analysis). This type of information would be very useful for later studies.

Despite the limitations, this research has several relevant points, not only in terms of its objective to compare information about patients from two different periods but also because it is the only study in the Spanish population allowing us to characterize the specific profile of patients admitted to a specialized Burns Unit due to SIB. However, the data come from a single unit, and despite this being a reference unit, it would be advisable to design longitudinal multicentric studies to continue assessing and characterizing this specific surviving population.

## 5. Conclusions

The main objective of this study was to compare data from two patient samples belonging to the same Burns Unit, under the same consistent admission criteria, but with a 20-year temporal gap. The data published in 1994 (Study I, 1983–1991) were compared with current data (Study II, 2010–2015), yielding results that indicate substantial changes in the profile of these patients over this period.

Some of the significant differences observed include the rate of admitted patients with a psychiatric background and the location of the incident (ranging from home to public spaces), as well as the type of triggers that lead to SIB, shifting from primarily psychopathological factors to a combination of psychopathological and psychosocial aspects. Concerning socio-economic health determinants, evidence from the scientific literature indicates a marked increase in external precipitators of SIB [24,26,32]. New professional profiles, such as social workers, could be indispensable additions to the healthcare team.

Our findings highlight important changes in the profile of patients with self-inflicted burns over time, suggesting an evolving interplay between psychiatric, psychosocial, and socio-economic factors. Future research should explore these dynamics in greater depth, particularly focusing on targeted interventions that address the unique needs of non-native populations and those experiencing socio-economic adversity. Additionally, longitudinal studies that assess the long-term psychosocial and physical sequelae in survivors of SIB could offer valuable insights to improve prevention and care strategies.

Finally, at the clinical level, as expected, due to the advances in healthcare, the time of hospitalization and mortality rates of these patients have been reduced. Concerning the survivors, questions are raised about their specific needs, the longitudinal evolution of their physical sequelae, and their psychopathological state and resolution after committing SIB.

## Figures and Tables

**Table 1 ebj-06-00008-t001:** Summary of socio-demographics and clinical characteristics of the sample.

Variables	Study I: 1983–1991 [6]	Study II: 2010–2015	
Duration of the study	9 years	5 years	
Number of patients	67	36	
Variables			*p*-value
Total percentage of admissions at the Burns Unit related to SIB	1.98%	1.45%	ns
Gender			ns
- Male	n = 48 (71.6%)	n = 28 (77.8%)	
- Female	n = 19 (28.4%)	n = 8 (22.2%)	
Age (in years)	Mean (SD) = 38 (14)	Mean (SD) = 44 years (14)	0.041
Marital status			ns
- Single, no partner	n = 41 (61.2%)	n = 25 (68.8%)	
- Married, with partner	n = 26 (38.8%)	n = 11 (31.3%)	
Place of the incident			0.020
- Home	n = 45 (67.2%)	n = 19 (52.8%)	
- Psychiatric hospital	n = 8 (12%)	n = 1 (2.8%)	
- Jail	n = 7 (10.4%)	n = 3 (8.3%)	
- Public arena	n = 7 (10.4%)	n = 12 (33.3%)	
- Other spaces	n = 0 (0%)	n = 1 (2.8%)	
Method to cause the burn			0.004
- Flame	n = 44 (65.7%)	n = 34 (94.4%)	
- Gas	n = 19 (28.3%)	n = 0 (0%)	
- Electrocution	n = 1 (1.5%)	n = 1 (2.8%)	
- Others	n = 3 (4.5%)	n = 1 (2.8%)	
Psychiatric history (yes)	n = 40 (75%)	n = 33 (91.7%)	0.001
Previous suicide attempts (yes)	n = 14 (20.9%)	n = 17 (47.2%)	0.007
Precipitant causes			0.036
- Psychopathology	n = 49 (73.1%)	n = 19 (52.8%)	
- Psychosocial	n = 14 (20.9%)	n = 16 (44.4%)	
- Others	n = 4 (6%)	n = 1 (2.8%)	
Total burnt area of the body			ns
- >50%	n = 19 (28.4%)	n = 9 (25%)	
- 30–50%	n = 15 (22.4%)	n = 4 (11.1%)	
- Between 10 and 30%	n = 23 (34.3%)	n = 11 (30.6%)	
- <10%	n = 10 (14.9%)	n = 12 (33.3%)	
Length of hospital stay (days)	Mean = 40.2 days	Mean = 32.31 days	ns
Mortality (yes)	n = 20 (29.8%)	n = 7 (19.4%)	ns

SIB: self-inflicted burns; TBA: total burn area; ns: nonsignificant.

## Data Availability

The data presented in this study are available on request from the corresponding author due to their sensitive nature and will only be shared with specific and justified demands, subject to review by the researchers and the institution’s ethical committee where the research was conducted.

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
