# Peer review of "Self-Inflicted Burns: A Comparative Study in a Spanish Sample"

_2673-1991, 2025, doi:10.3390/ebj6010008_

Round 1

Reviewer 1 Report (Previous Reviewer 2)

Comments and Suggestions for Authors

The manuscript is relevantly better and easier to read than earlier. The references add also relevant and better basis for this study.

However, concerning the terminology of self-destructive and suicidal behavior I would like to have more exact definitions in the methods or earlier in the text.

SIB (=self-inflicted burn) definition in detail in Garcia-Sanchez (study I) and in present study (study II). I do not find the clear and open definition. Does it include both self-harm burns without an intention to kill oneself and also attempted suicide by burn with intention to kill oneself. I could not have access to Garcia´s full text anymore online and ask also if the definition was exactly the same. This is very relevant for the entire study design.

Another term which needs more exact definition autolytic attempt. Even if I have been active in this field of suicide research about 30 years I never met this word! It needs clear and a exact definition.

Author Response

Thank you for review and pointing our article. We agree with your comment .

About clear terminology: “I would like to have more exact definitions in the methods or earlier in the text. SIB (=self-inflicted burn) definition in detail in Garcia-Sanchez (study I) and in present study (study II).”

Regarding the concept of SIB (self-inflicted burns), there are divergences between authors, depending on their healthcare specialties and also on cultural contexts. In our Burns Unit, we refer to SIB as patients who come to our emergency department and are then admitted, when the etiology of the wounds is self-inflicted. This aspect is central, since research points to the slowest progress and the severity of the patients. Historically in Spain, the great majority of SIBs had very serious mental pathologies, and as this comparative study shows, SIBs, since the 2010s, turn out to be patients who carry out these acts for reasons that we describe as psychosocial.

Another term which needs more exact definition autolytic attempt. Even if I have been active in this field of suicide research about 30 years I never met this word! It needs clear and a exact definition.

In mental health specialties, such as psychiatry and clinical psychology, the concept of suicidal intent refers to a person's intention to commit an act that threatens his or her life. Suicidal attempts manifest themselves in various ways, but they are important when they are analyzed in terms of their intentionality. For example, when patients commit these acts with high lethality and low salvageability, versus patients who commit them with high lethality and low salvageability.

Reviewer 2 Report (Previous Reviewer 3)

Comments and Suggestions for Authors

No further comment provided. 

Author Response

Thank you for review and pointing our article. 

Reviewer 3 Report (Previous Reviewer 4)

Comments and Suggestions for Authors

Thank you for allowing me to review your manuscript.

1. An important topic, as a stronger focus on mental health has been a mainstay in burn care for several decades now. I wonder if there are deeper conclusions to be drawn from this topic and paper, which could be added in a next steps discussion. However, this is not required.

2. It is unclear to me the importance of 'foreign' status. I will not infer what the connection may be in the authors' minds, but I think a robust discussion is necessary if this is to be mentioned.

Author Response

Thank you for review and pointing our article. We agree with your comment .

Comment 1 An important topic, as a stronger focus on mental health has been a mainstay in burn care for several decades now. I wonder if there are deeper conclusions to be drawn from this topic and paper, which could be added in a next steps discussion. However, this is not required.

This comment is very interesting, because in the Hospitalization Units, we are aware of the changes in the profiles of the patients and the health and social needs they require. In this increase of patients with psychosocial causes for their SIB, the presence of social workers is providing resources and perspectives for their recovery and social integration.

The effort to standardize this care and the inclusion of perspectives beyond the surgical ones is being very relevant.

Comment 2. It is unclear to me the importance of 'foreign' status. I will not infer what the connection may be in the authors' minds, but I think a robust discussion is necessary if this is to be mentioned.

Thank you for allowing us to clarify this concept of "foreign" status. Of course, this is not a value judgment of any kind about our patients, but in the comparative study we point out the contextual and social changes in the years following the first study. Spain, which had practically no migrant population, went on to significantly increase its population by nearly 15% thanks to the arrival of people from other parts of the world. This is the intention of pointing out the presence of foreign people, or in other words, non-natives.

This manuscript is a resubmission of an earlier submission. The following is a list of the peer review reports and author responses from that submission.

Round 1

Reviewer 1 Report

Comments and Suggestions for Authors

Very interesting comparison of studies I and II.  Findings reflect social changes as well as increased survival due to advancements in burn treatment.

Comments on the Quality of English Language

There is a minor need of editing for English syntax but otherwise the manuscript is very well written.

Author Response

Thank you very much for your review, focused on the importance of our findings, and for the suggestions about improving language expression

Reviewer 2 Report

Comments and Suggestions for Authors

Reviewer comments on Manuscript ebj-2515278: 

This manuscript “Self-inflicted burns: A comparative study in a Spanish sample raises an important phenomena which has significant effects/impact on burn patients in intensive care and in standard burn ward: Self-inflicted burns (SIB). 

The objective of this study was to compare data of patients admitted for SIBs in a specialized Burn Unit in Spain under the same admission criteria in two different time periods. (Between 1983-1991 in Study I, and between 2010-2015 in Study II). Substantial changes in the profile of these patients in a later period were focused on the data. The study tries also to discuss the possibility of these relationships with the relevant socio-cultural and economic changes that have occurred through the time-period. Methodologically, a replication of the initial design was performed in Study II, including in the analysis only the variables considered in Study I. 

All this has been processed in very proper and professional way. 

However, I want to point out a couple of things and suggestions for changes which could help the reader:

Generally: I find Introduction, Discussion and Conclusion very thorough and in-depth which is good but long. This can make it heavy and difficult for the reader. However, here I find perhaps too much repetition. 

 Introduction (80 lines): It has a very good review of earlier studies focusing on SIB, of which many are familiar to me. Last chapter of Introduction, I am used here to describe very clearly, shortly and purely the aims of the present study. 

Materials and Methods (45 lines) :

Setting: “This Burn Unit is currently receiving patients from an area of influence of more than 8.500.000 patients, included local population, Balears Island and Andorra Principat also”. This is difficult to believe. Could the sentence be formulated more clearly. Furthermore, should it be instead, 8.500.000 people/ citizens? Besides, the Methods seem to me ok. 

 Results (text 35 lines):

The results seem to me ok. Compact and keeping closely enough and just focused into the results of this study. 

Discussion (129 lines):

Should the first chapter not have in short, the core findings of this study and the chapter comparing present findings to those in earlier studies follow later? 

Should the limitations of the study to be discussed more compactly and in discussion? 

Conclusion (s) (48 lines): Not “Conclussions”. I am used to shorter Conclusion part with just the very few sentences about conclusions which are based on this study and lift the results on an more general level and give perhaps some clinical implications or recommendations for future research.

Furthermore, shouldn´t the strengths and limitations of the study be in discussion? 

Table 1: I recommend that Table 1 would be close by the Results, as in this manuscript, also in the final layout of the Journal.

Final comment: I recommend that these comments above should be considered before this manuscript would be published.

Comments on the Quality of English Language

Minor editing needed.

Reviewer 3 Report

Comments and Suggestions for Authors

Thank you for allowing me to review this study entitled: Self-inflicted burns: A comparative study in a Spanish sample. 

The authors haver not conducted a literature review as many key articles have not been incorporated into this study. I strongly suggest the authors reflect the currency of the literature in the introduction and discussion.

Please review grammar and spelling e.g. CONCLUSSIONS

Comments on the Quality of English Language

Please review grammar and spelling e.g. CONCLUSSIONS

Author Response

First of all, thank you about your rich review of our article.

About the specific aspects of lingüistic limitations, we are waiting for the bilingual english-spanish translator, for a hole review of the material. In the next days, we are going to send you again the text, with the corrections of the contents and the improve of the language, gramma and spelling.

About specifical aspects:

The authors haver not conducted a literature review as many key articles have not been incorporated into this study. I strongly suggest the authors reflect the currency of the literature in the introduction and discussion.

We are just solving this commentary, just reviewing again the scientific literature about this specifical issue. 

Reviewer 4 Report

Comments and Suggestions for Authors

Thank you for letting me review this manuscript.

A very important topic that needs to be examined and an important dialogue to be had in the burn community.

1. Some of the grammar is off and perhaps some of the phrasing is unique to country of origin or Europe. I assume autolysis means suicide attempt, which makes sense, but I did have to go look it up and it did nag at me as I read through the paper. A minor distraction that I don't think is necessary.

2. Would be interesting to see some disposition data. Did these patients end up going to rehab or a nursing facility? Did many of them go to psychiatric facilities after? Important to discuss.

3. Would be nice for an outsider to receive some context about mental health resources in the country of origin. Is it easy to access? Do you need to have appropriate insurance? Do you need a referral?

4. Has anything changed in terms of access or first responders or anything that would explain the changes in demographics from study 1 to 2? If this can be elucidated, then perhaps action can be taken to further improve outcomes.

5. How did the authors choose the second time period?

Overall, a very important paper. Would love for some more context and some editing to grammar/spelling, which would simply help the readability and distractions in the paper currently.

Comments on the Quality of English Language

Some spelling errors and differences in terms from American English, at least. Autolysis is confusing for me, but understandable. Small things like 'burns units' reads strangely.

Overall understandable, but an unnecessary distraction from an important topic.

Author Response

First of all, thank you very much for your deeply review of our article. All the comments are very welcoming.

About:

 1.-Would love for some more context and some editing to grammar/spelling, which would simply help the readability and distractions in the paper currently.

Taking in account the reviwers suggestions, we are introducing them and sending the article for a specialist translator.

2. Would be interesting to see some disposition data. Did these patients end up going to rehab or a nursing facility? Did many of them go to psychiatric facilities after? Important to discuss.

Yes, these patients receive their psychiatric attention outside the Burn Unit and start later and during this phase, the Rehabilitation, as the other burn patients. This aspecte we'll incorporate in the text for the next revision

3. Would be nice for an outsider to receive some context about mental health resources in the country of origin. Is it easy to access? Do you need to have appropriate insurance? Do you need a referral?

As Mental Helath Services are organizate in Catalonia, all the patients with psychiatric needs are derivated to their specific Units, close to their regions of living. So, we can be in contact with these new teams, to help the recovery of the SIB patients

4. Has anything changed in terms of access or first responders or anything that would explain the changes in demographics from study 1 to 2? If this can be elucidated, then perhaps action can be taken to further improve outcomes.

Yes, new social and economicaly conditions could be principal factors in SIB episodes in Study 2. Part of them we can consider in the Rehabilitation phase, adding to psychiatric factors, more frequents in the first strudy.

5. How did the authors choose the second time period?

Good questions, because it is related to the first period electronical records in our Burn Unit. It allow us to find the data and the personal information more extended than in the first Study. Intersting aspect to add in the reviewed article
